# Physics of Language Models: Part 3.3, Knowledge Capacity Scaling Laws

## [Extended Abstract]*

**Zeyuan Allen-Zhu**
FAIR at Meta
zeyuanallenzhu@meta.com

**Yuanzhi Li**
Mohamed bin Zayed University of AI
Yuanzhi.Li@mbzuai.ac.ae

## Abstract

Scaling laws describe the relationship between the size of language models and their capabilities. Unlike prior studies that evaluate a model's capability via loss or benchmarks, we estimate information-theoretically the number of knowledge *bits* a model stores. We focus on factual knowledge represented as tuples, such as (USA, capital, Washington D.C.) from a Wikipedia page. Through multiple controlled datasets, we establish that language models can and only can store *2 bits of knowledge per parameter, even when quantized to int8*, and such knowledge can be flexibly extracted for downstream applications. *More broadly, we present 12 results* on how (1) training duration, (2) model architecture, (3) quantization, (4) sparsity constraints such as MoE, and (5) data signal-to-noise ratio affect a model's knowledge storage capacity.

## 1 Introduction

The scaling laws of large language models remain a pivotal area of research, enabling predictions about the performance of extremely large models through experiments with smaller ones. On the training time aspect, established scaling laws (Hoffmann et al., 2022; Kaplan et al., 2020; Hernandez et al., 2021; Alabdulmohsin et al., 2022; Henighan et al., 2020) discuss the optimal training flops versus model size. However, recent studies (Muennighoff et al., 2023; Gunasekar et al., 2023; Li et al., 2023) challenge these laws, demonstrating that training smaller models with significantly more flops can yield superior results. While these laws talk about how much time/data is needed to train a model of a certain size, another fundamental question is: *what is the ultimate performance a model can achieve, assuming sufficient training*? Despite the known emergent behaviors in large models (Bubeck et al., 2023; Yu et al., 2023), or even qualitative arguments that modern large models have reached L2 or L3-level intelligence (Allen-Zhu & Xu, 2025), there is a *lack of a principled, quantitative analysis* on how model size impacts its capacity when adequately trained.[1]

Traditional theory on overparameterization suggests that scaling up model size in sufficiently trained models can enhance memorization of training data (Allen-Zhu et al., 2019b), improve generalization

---

*This paper is part of the *Physics of Language Models* series, one of the first six papers presented as a two-hour tutorial at ICML 2024 in Austria (youtu.be/yBL7J0kgldU). Full and future editions of Part 3.3, including additional experiments and potential code releases, are available at physics.allen-zhu.com and ssrn.com/abstract=5250617.

[1]There is a rich literature comparing how pretrained models perform on benchmark tasks. Most comparisons are for different model families trained over different data: if LLaMA-70B is better than Mistral-7B, does the gain come from its choice of pretrain data, or the architecture difference, or really the size of the model? Some comparisons are among the same architecture, such as LLaMA-70B scores 63.6% on the world knowledge benchmark while LLaMA-7B scores only 48.9% (Touvron et al., 2023b); does this mean increasing model size by 10x increases its capacity only to $130\% = 63.6/48.9$? Thus, it is highly important to use a more principled framework to study scaling laws in a controlled setting.

error (Hestness et al., 2017; Rosenfeld, 2021; Rosenfeld et al., 2019), and better fit complex target functions (Li & Liang, 2018; Allen-Zhu et al., 2019a). However, these results often overlook large constant or polynomial factors, leading to a significant discrepancy from practical outcomes.

In this paper, we introduce a principled framework to examine *highly accurate* scaling laws concerning model size versus its *knowledge storage capacity*. It is intuitive that larger language models can store more knowledge, but does the total knowledge scale linearly with the model's size? What is the **exact constant** of this scaling? Understanding this constant is crucial for assessing the efficiency of transformer models in knowledge storage and how various factors (e.g., architecture, quantization, training duration, etc.) influence this capacity.

Knowledge is a, if not the, pivotal component of human intelligence, accumulated over our extensive history. Large language models like GPT-4 are celebrated not just for their sophisticated logic but also for their superior knowledge base. Despite rumors of GPT-4 having over 1T parameters, *is it necessary to store all human knowledge?* Could a 10B model, if trained sufficiently with high-quality data, match GPT-4's knowledge capacity? Our paper seeks to address these questions.

**Knowledge Pieces.** Defining "one piece of human knowledge" precisely is challenging. This paper aims to make progress by focusing on a restricted, yet sufficiently interesting domain. We define a *piece* of knowledge as a (name, attribute, value) tuple, e.g., (Anya Forger, birthday, 10/2/1996); and many data in world knowledge benchmarks can be broken down into pieces like this.[2]

We generate *synthetic* knowledge-only datasets by uniformly at random generating (name, attribute, value) tuples from a knowledge base and converting them into English descriptions. We pretrain language models (e.g., GPT-2, LLaMA, Mistral) on these texts using a standard auto-regressive objective from random initialization, and "estimate" the learned knowledge. By varying the number of knowledge pieces and model sizes, we outline a knowledge capacity scaling law.

Our idealized setting, free from irrelevant data, allows for more accurate scaling law computations — we also discuss how "junk" data affects capacity. In contrast, it is difficult to quantify real-life knowledge; for instance, if LLaMA-70B outperforms LLaMA-7B by 30% on a benchmark, it doesn't necessarily mean a tenfold model scaling only boosts capacity by 30% (see Footnote 1). The synthetic setting also lets us adjust various hyperparameters, like name/value lengths and vocabulary size, to study their effects on knowledge capacity scaling laws.

Most of the paper shall focus on a setting with synthetically-generated human biographies as data, either using predefined sentence templates or LLaMA2-generated biographies for realism.

**Bit Complexity and Capacity Ratio.** For $N$ knowledge pieces (i.e., $N$ tuples), we define the *bit complexity* as the minimum bits required to encode these tuples. For any language model trained on this data, we calculate its "bit complexity lower bound" (see Theorem 3.1), describing the minimum number of bits needed for the model to store the knowledge at its given accuracy. This formula is nearly as precise as the upper bound, within a $1 - o(1)$ factor.

We train language models of varying sizes on knowledge data with different $N$ values. By comparing the models' trainable parameters to the bit complexity lower bounds, we evaluate their knowledge storage efficiency. A model with 100M parameters storing 220M bits of knowledge has a *capacity ratio* of 2.2 bits per parameter.

**Our results.** Our findings are summarized as follows:

- RESULTS 1-3: BASE SCALING LAW FOR GPT2. [3]
  - RESULT 1+2+3: GPT2, trained with standard AdamW, consistently achieves a 2bit/param capacity ratio across all data settings after sufficient training. This includes various model sizes, depths, widths, data sizes, types (synthetic/semi-synthetic), and hyperparameters (e.g., name/value length, attribute number, value diversity).

---

[2]Examples include (Africa, largest country, Sudan) and (It Happened One Night, director, Frank Capra) in TriviaQA (Joshi et al., 2017), or (Teton Dam, collapse date, 06/05/1976) and (USA, Capital, Washington D.C.) in NaturalQuestions (Kwiatkowski et al., 2019).

[3]In this paper, GPT2 refers to that the GPT2 model with rotary embedding instead of positional embedding and without dropout.

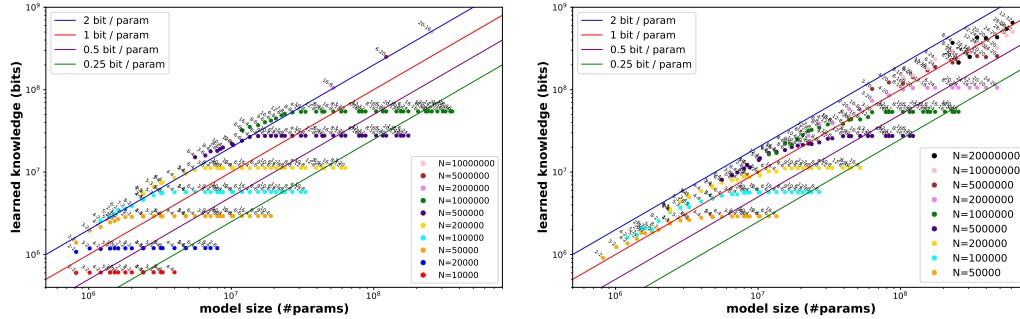

(a) bioS($N$) data (**1000 exposures**), peak $R(F) \geq 2$  (b) bioS($N$) data (**100 exposures**), peak $R(F) \geq 1$

Figure 1: Scaling laws for GPT2 pretrained on bioS($N$) data using fp16 (mixed-precision) for 1000/100 exposures.

---

**Conclusion.** The *peak* capacity ratios consistently exceed $R(F) \geq 2$ (resp. $\geq 1$) for 1000 exposures (resp. 100 exposures) of pretraining on each knowledge piece, **regardless of model depth/size**.

---

**Remarks.** Each dot $\ell$-$h$ represents GPT2 with $\ell$ layers, $h$ heads, and $64d$ dimensions. The learned knowledge is calculated by the bit-complexity lower bound Theorem 3.1. The full paper also includes: similar results for bioS$^{\text{simple}}(N)$ and bioR($N$) data, the *same holds* for quantization using int8, and confirming full extractability of all learned knowledge.[5]

---

**Larger models?** Training GPT2-20-16 on bioS($10M$) for 1000 exposures costs 8.5 days with 64 A100s, while GPT2-12-32 on bioS($20M$) for 100 exposures took 2.4 days. In our synthetic setting, we see no need to scale up further. Instead, we prefer to allocate GPUs to explore other aspects covered in this paper.

---

*Remark* 1.1. This predicts **a sufficiently trained 7B language model** can store 14B bits of knowledge, surpassing the knowledge of English Wikipedia and textbooks by our estimation.[4]

*Remark* 1.2. When we say the model *stores knowledge*, it isn't word-by-word memorization. Instead, the knowledge is flexibly extractable (e.g., via QAs like "What is Anya Forger's birthday") (Allen-Zhu & Li, 2024) and applicable in downstream tasks (e.g., comparing birthdays) via fine-tune (Allen-Zhu & Li, 2025).

- RESULT 4: HOW TRAINING TIME AFFECTS MODEL CAPACITY.

  Achieving a 2bit/param capacity requires each knowledge piece to be visited 1000 times during training, termed ***1000-exposure*** to differentiate from traditional "1000-pass" terminology, as a single data pass can expose a knowledge piece 1000 times.[6]

  - RESULT 4: With 100 exposures, an *undertrained* GPT2's capacity ratio falls to 1bit/param. (See Figure 1.)

  *Remark* 1.3. Another perspective on Result 4 is that *rare* knowledge, encountered only 100 times during training, is stored at a 1bit/param ratio.

- RESULTS 5-7: HOW MODEL ARCHITECTURE AFFECTS MODEL CAPACITY.

  We tested LLaMA, Mistral, and GPT2 architectures with reduced or even no MLP layers.

---

[4]As of February 1, 2024, English Wikipedia contains a total of 4.5 billion words, see `https://en.wikipedia.org/wiki/Wikipedia:Size_of_Wikipedia#Size_of_the_English_Wikipedia_database`, accessed March 2024. We estimate that the non-overlapping contents of English textbooks have fewer than 16 billion words in total, see Remark P.1. This amounts to 20.5 billion words, and we believe they contain fewer than 14 billion bits of knowledge.

[5]A distinction exists between memorizable knowledge (e.g., text memorized during pretraining) and knowledge flexibly extractable via instruction fine-tuning (Allen-Zhu & Li, 2024); our results in this paper apply to both.

[6]For example, it is plausible that one pass through Wiki data might present the knowledge piece (US, capital, Washington D.C.) 1000 times, and one pass through the Common Crawl might present it a million times.

- RESULT 5: In the 1000-exposure setting, a 2bit/param capacity ratio appears to be a **universal rule**: all models, even without MLP layers, closely achieve this ratio.
- RESULT 6: With 100 exposures, some archs show limitations; notably, LLaMA/Mistral's capacity ratio is 1.3x lower than GPT2's, even after best-tuned learning rates.
- RESULT 7: Further controlled experiments indicate that "gated MLP" usage leads to LLaMA/Mistral architecture's underperformance in knowledge storage.

*Remark* 1.4. **Our framework offers a principled playground to compare models.** This contrasts with traditional comparisons based on loss/perplexity, which can produce debatable conclusions.[7] Controlled data also reveal more significant differences between models.[8]

- RESULT 8: HOW QUANTIZATION AFFECTS MODEL CAPACITY.

We applied GPTQ (Frantar et al., 2022) to quantize models from the base scaling laws to int8 or int4. Surprisingly,

- RESULT 8: Quantizing to int8 does not compromise model capacity (even for models on the boundary of 2bit/param); however, quantizing to int4 reduces capacity to 0.7bit/param.

*Remark* 1.5. Since int8 is 8bit, LLMs can exceed 1/4 of the theoretical limit for storing knowledge; thus knowledge must be very compactly stored inside the model across all layers.

*Remark* 1.6. Since 2bit/param is obtained after sufficient training, training longer *may not* further improve model capacity, *but quantization can*. While not covered in this paper, our framework also provides a principled playground to compare different quantization methods.

- RESULT 9: HOW SPARSITY (MOE) AFFECTS MODEL CAPACITY.

Mixture-of-experts (MoE) models offer faster inference than dense models but often underperform dense models with the same total parameter count (not effective parameters). We show that this performance drop is likely not due to a lack of knowledge storage capability.

- RESULT 9: MoE models, even with 32 experts, only reduce 1.3x in capacity compared to the base scaling laws, despite using just $8.8\%$ of the total parameters during inference.

- RESULTS 10-12: HOW JUNK KNOWLEDGE AFFECTS MODEL CAPACITY.

Not all pretrain data are equally useful. Much of the internet data lacks valuable knowledge for training language models (Li et al., 2023), while knowledge-rich sources like Wikipedia represent only a small fraction of the training tokens. We explore the impact on model capacity by conducting a controlled experiment with both useful and "junk" data.

- RESULT 10+11: Junk data significantly reduces model capacity. As an example, with a 1:7 ratio of "useful to junk" training tokens, capacity for useful knowledge *loses by a factor of 20*x, even when useful knowledge is exposed 100 times.[9]
- RESULT 12: An *effective mitigation* is to prepend a special token to all useful knowledge. This is akin to adding a domain name like wikipedia.org at the start of every Wikipedia paragraph; the model *autonomously* identifies high-quality data without prior knowledge of valuable domains. In the example above, the loss factor improves from 20x to 2x.

**Conclusion.** Overall, our approach to studying knowledge capacity scaling laws offers a flexible and **more accurate playground** compared to traditional methods that evaluate language models trained on internet data against real-world benchmarks. This accuracy is partly due to the synthetic nature of our dataset, which eliminates concerns such as data contamination that could compromise the validity of real-world benchmark results. In this paper, we've conducted a thorough comparison across different model architectures and types of knowledge. While we haven't explored various quantization methods, this represents a promising direction for future research. We've also investigated the impact of junk data and proposed mitigation strategies. We believe the insights gained from this principled exploration can assist practitioners in making informed decisions about model selection, training data preparation, and further theoretical research into LLMs.

---

[7]A model might achieve better perplexity by performing *much better* on simpler data but poorer on complex data, or by excelling in reasoning but not in knowledge. Our results offer a more nuanced view: GatedMLP doesn't affect frequent knowledge but does impact moderately rare knowledge (e.g., with 100 exposures).

[8]For example, Shazeer (2020) found GatedMLP offers a $\sim 1\%$ accuracy boost on benchmark tasks; our findings of a 1.3x difference translates for instance to accuracies 90% vs. 70%.

[9]The loss factor improves to 3x/1.5x/1.3x with 300/600/1000 exposures of useful knowledge, compared to Result 4 which involves training without junk for only 100 exposures.

## 2 PRELIMINARIES

In this paper, a piece of knowledge is a tuple of three strings: (name, attribute, value) $= (n, a, v)$. For instance, $n =$ "Anya", $a =$ "birthday", $v =$ "Oct 2, 1996".

### 2.1 KNOWLEDGE (THEORETICAL SETTING)

The complexity of a knowledge set is determined not only by the number of knowledge pieces but also by the length of the value string $v$, the diversity of the vocabulary, and other factors. For instance, if the attribute $a =$ "passport number," then the value $v$ contains more bits of knowledge compared with $a =$ "gender," because the former has significantly higher *diversity*. If the attribute $a =$ "birth date," then the value $v$ could consist of 3 *chunks*: $(10, 2, 1996)$.

Considering these examples, we propose a set of hyperparameters that may influence the complexity of knowledge:

1. $N$ — the number of (distinct) names $n$, denoted by $\mathcal{N}$.
2. $K$ — the number of attributes $a$, with $\mathcal{A}$ representing the set of attributes. For simplicity, we assume $|\mathcal{A}| = K$ is fixed.
3. $T$ — the number of tokens $T$, where every character in $v$ belongs to $\mathcal{T}$ for some $|\mathcal{T}| = T$. For example, we can think of $T$ as "vocab size" in a tokenizer.
4. $C$ and $L$ — the number of chunks and the length of each chunk for the value: each value $v \in (\mathcal{T}^L)^C$ can be expressed as $v = (v_1, v_2, \cdots, v_C)$, where $v_i \in \mathcal{T}^L$.
5. $D$ — the diversity of chunks: for each piece of knowledge $(n, a, v)$ and $i \in [C]$, the chunk $v_i$ belongs to $\mathcal{D}_a \subset \mathcal{T}^L$, for some set with cardinality $D \overset{\text{def}}{=} |\mathcal{D}_a| \ll T^L$.

*Remark* 2.1. For notation simplicity, we have assumed that all chunks within an attribute $a \in \mathcal{A}$ share the same diversity set $\mathcal{D}_a$, and all chunks are of equal length, etc. This enables us to more easily demonstrate the influence of each hyperparameter on a model's capacity. In practice, different attributes may have different diversity sets or value lengths — e.g., $\mathcal{D}_{\text{passport}}$ could be much larger than $\mathcal{D}_{\text{gender}}$. Our theoretical results do apply to these settings, albeit with more complex notation.

In our theoretical result, we introduce a dataset $\mathsf{bioD}(N, K, C, D, L, T)$ defined as follows:

**Definition 2.2** ($\mathsf{bioD}$ data generation). *Consider a fixed set of $K$ attributes, such as a set $\mathcal{A} = \{$"ID 1" $\ldots$ "ID K"$\}$, and a fixed set $\mathcal{N}_0$ of candidate names (with $N_0 \overset{\text{def}}{=} |\mathcal{N}_0| \gg N$).*

1. *Generate $N$ names uniformly at random (without replacement) from $\mathcal{N}_0$ to form $\mathcal{N}$.*
2. *For each attribute $a \in \mathcal{A}$, generate $D$ distinct strings $w_{1,a}, \cdots, w_{D,a} \in \mathcal{T}^L$ uniformly at random (without replacement) to form the diversity set $\mathcal{D}_a$.*
3. *For each name $n \in \mathcal{N}$ and attribute $a \in \mathcal{A}$, generate value $v^\star(n, a) = (v_1, v_2, \cdots, v_C)$ by sampling each $v_i \in \mathcal{D}_a$ uniformly at random.*

*Let $\mathcal{Z} \overset{\text{def}}{=} \{(n, a, v^\star(n, a))\}_{n \in \mathcal{N}, a \in \mathcal{A}}$ be the knowledge set.*

**Proposition 2.3** (trivial, bit complexity upper bound). *Given $\mathcal{N}_0$ and $\mathcal{A}$ and $\mathcal{T}$, to describe a knowledge set generated in Definition 2.2, one needs at most the following number of bits:*

$$\log_2 \binom{|\mathcal{N}_0|}{N} + NKC \log_2 D + K \log_2 \binom{T^L}{D} \approx N \log_2 \frac{|\mathcal{N}_0|}{N} + NKC \log_2 D + KD \log_2 \frac{T^L}{D} \ .$$

(The approximation is valid when $|\mathcal{N}_0| \gg N$ and $T^L \gg D$.) We will present a bit complexity lower bound in Section 3.

### 2.2 KNOWLEDGE (EMPIRICAL SETTING)

We utilize both the synthetic $\mathsf{bioD}$ dataset, generated as per Definition 2.2, and several human biography datasets to evaluate language model scaling laws.

Allen-Zhu & Li (2024) introduced a synthetic biography dataset comprising $N$ randomly-generated (fake) individuals, each characterized by six attributes: birth date, birth city, university, major, em-

ployer, and working city.[10] To translate these tuples into natural language, in their bioS dataset, each individual is described by six randomly selected English sentence templates corresponding to their attributes. We direct readers to their paper for more details but provide an illustration below:

Anya Briar Forger was born on October 2, 1996. She spent her early years in Princeton, NJ. She received mentorship and guidance from faculty members at Massachusetts Institute of Technology. She completed her education with a focus on Communications. She had a professional role at Meta Platforms. She was employed in Menlo Park, CA.

$$(2.1)$$

In this paper, we explore three variations of such datasets:

- bioS($N$) represents an online dataset for $N$ individuals, where each biography is generated with new randomness for the *selection* and *ordering* of six sentence templates *on-the-fly*.

- bioS$^{\text{simple}}$($N$) denotes a similar dataset, but here, each biography is generated once with a fixed random selection and ordering of the sentence templates.

- bioR($N$) refers to the same dataset, but with each biography written 40 times by LLaMA2 (Touvron et al., 2023b) to increase realism and diversity.

These datasets correspond to the bioS multi+permute, bioS single+permute, and bioR multi data types discussed in (Allen-Zhu & Li, 2024), albeit with minor differences. While their study focused on $N = 100K$, we expand our scope for bioS to consider $N$ up to $20M$; for bioR, we limit $N$ to $1M$, which already yields a dataset size of 22GB.

As introduced in Section 1, if each knowledge piece is seen 1000 times during training, we call this ***1000 exposures***. For bioS($N$), 1000 exposures will unlikely include identical biography data because there are 50 sentence templates for each attribute and a total of $50^6 \times 6!$ possible biographies per person. For bioS$^{\text{simple}}$($N$), 1000 exposures mean 1000 passes of the data. For bioR($N$), 1000/100 exposures mean only 25/2.5 passes of the training data.

For the bioD dataset, we define $\mathcal{N}_0$ to be identical to bioS, with $|\mathcal{N}_0| = 400 \times 400 \times 1000$. We encapsulate a person's attributes within a single paragraph, employing random sentence orderings and a consistent sentence template. For example:

Anya Briar Forger's ID 7 is $v_{7,1}, \ldots, v_{7,C}$. Her ID 2 is $v_{2,1}, \ldots, v_{2,C}$. [...] Her ID 5 is $v_{5,1}, \ldots, v_{5,C}$.

In this paper, we primarily utilize bioS. To illustrate broader applicability and *to better connect to theoretical bounds*, we also present results for bioS$^{\text{simple}}$, bioR, and bioD.

### 2.3 MODELS AND TRAINING

GPT2 was introduced in (Radford et al., 2019). Due to its limitations from the absolute positional embedding (Allen-Zhu & Li, 2023), we adopt its *rotary positional embedding* variant (Su et al., 2021; Black et al., 2022), which we still refer to as GPT2 for convenience. Additionally, we disable dropout, which has been shown to improve performance in language models (Touvron et al., 2023b). We explore a wide range of model sizes while using a fixed dimension-per-head of 64. The notation GPT2-$\ell$-$h$ represents $\ell$ layers, $h$ heads, and $64h$ dimensions; for example, GPT2-small corresponds to GPT2-12-12. The default GPT2Tokenizer is used, converting people's names and most attributes into tokens of variable lengths. In examining the impact of model architectures on scaling laws, we will also use LLaMA/Mistral architectures (Touvron et al., 2023a; Jiang et al., 2023).

**Training.** We train language models *from scratch (i.e., random initialization)* using the specified datasets. Knowledge paragraphs about individuals are randomly concatenated, separated by <EOS> tokens, and then randomly segmented into 512-token windows. The standard autoregressive loss is employed for training. Unless specified otherwise, training utilizes the default AdamW optimizer and mixed-precision fp16. Learning rates and weight decays are moderately tuned (see full paper).

---

[10]All attributes, except for the working city (determined by the employer's headquarters), are chosen uniformly and independently at random. There are $N_0 = 400 \times 400 \times 1000$ possible person names, $12 \times 28 \times 200$ birth dates, 200 birth cities, 300 universities, 100 majors, and 263 employers. Additionally, a random pronoun with 2 possibilities is chosen for each person.

## 3 BIT COMPLEXITY LOWER BOUND

When assessing the knowledge stored in a model, we **cannot** simply rely on the **average, word-by-word** cross-entropy loss. For example, the phrase "received mentorship and guidance from faculty members" in (2.1) does not constitute useful knowledge. We should instead focus on the *sum* of the loss for *exactly* the knowledge tokens.

Consider a model $F$ with weight parameters $W \in \mathcal{W}$. Assume $F$ is trained on a $\mathsf{bioD}(N, K, C, D, L, T)$ dataset $\mathcal{Z}$ as defined in Definition 2.2 using any optimizer; this process is represented as $W = W(\mathcal{Z})$ (the model's weight is trained as a function of the training dataset $\mathcal{Z}$). During the evaluation phase, we express $F$ through two functions: $F^\top(W, R)$, which generates names, and $F^\perp(W, n, a, R)$, which generates values given $(n, a)$, where $R$ denotes the randomness used in generation. Let $F_1^\perp(W(\mathcal{Z}), n, a, R)$ represent the first chunk of $F^\perp(W(\mathcal{Z}), n, a, R)$. We evaluate $F$ by calculating the following three cross-entropy losses:[11]

$$\mathbf{loss}_{name}(\mathcal{Z}) \stackrel{\text{def}}{=} \mathbb{E}_{n \in \mathcal{N}} - \log \mathbf{Pr}_R \left[ F^\top(W(\mathcal{Z}), R) = n \right]$$

$$\mathbf{loss}_{value1}(\mathcal{Z}) \stackrel{\text{def}}{=} \mathbb{E}_{n \in \mathcal{N}, a \in \mathcal{A}} - \log \mathbf{Pr}_R \left[ F_1^\top(W(\mathcal{Z}), n, a, R) = v_1^\star(n, a) \right]$$

$$\mathbf{loss}_{value}(\mathcal{Z}) \stackrel{\text{def}}{=} \mathbb{E}_{n \in \mathcal{N}, a \in \mathcal{A}} - \log \mathbf{Pr}_R \left[ F^\perp(W(\mathcal{Z}), n, a, R) = v^\star(n, a) \right]$$

We shall explain in the full paper that these quantities are easy to be derived from the auto-regressive entropy-loss using examples, and below we quickly state our bit-complexity lower bound theorem:

**Theorem 3.1** (bit complexity lower bound). *Suppose $N \geq \Omega(D \log N)$. We have*

$$\log_2 |\mathcal{W}| \geq \mathbb{E}_{\mathcal{Z}} \left[ N \log_2 \frac{N_0 - N}{e^{\mathbf{loss}_{name}(\mathcal{Z})}} + NK \log_2 \frac{D^C}{e^{\mathbf{loss}_{value}(\mathcal{Z})}} \right.$$
$$\left. + KD \log_2 \frac{T^L - D}{De^{(1+o(1))\mathbf{loss}_{value1}(\mathcal{Z})}} - o(KD) \right]$$
$$= N \log_2 \frac{N_0 - N}{e^{\mathbb{E}_{\mathcal{Z}} \mathbf{loss}_{name}(\mathcal{Z})}} + NK \log_2 \frac{D^C}{e^{\mathbb{E}_{\mathcal{Z}} \mathbf{loss}_{value}(\mathcal{Z})}}$$
$$+ KD \log_2 \frac{T^L - D}{De^{(1+o(1))\mathbb{E}_{\mathcal{Z}} \mathbf{loss}_{value1}(\mathcal{Z})}} - o(KD)$$

The goal of the paper is to study how the number of model parameters **competes with** this bound. We defer the proof to the full paper, and shall explain over there why proving such bound is non-trivial.

## 4 CAPACITY RATIO

Motivated by Theorem 3.1, ignoring lower order terms, we define the empirical capacity ratio as

**Definition 4.1.** *Given a model $F$ with $P$ parameters trained over a $\mathsf{bioD}(N, K, C, D, L, T)$ dataset $\mathcal{Z}$, suppose it gives $p_1 = \mathbf{loss}_{name}(\mathcal{Z})$, $p_2 = \mathbf{loss}_{value}(\mathcal{Z})$, $p_3 = \mathbf{loss}_{value1}(\mathcal{Z})$, we define its* capacity ratio *and* max capacity ratio

$$R(F) \stackrel{\text{def}}{=} \frac{N \log_2 \frac{N_0}{e^{p_1}} + NK \log_2 \frac{D^C}{e^{p_2}} + KD \log_2 \frac{T^L}{De^{p_3}}}{P} .$$

$$R^{\mathsf{max}}(F) \stackrel{\text{def}}{=} \frac{N \log_2 \frac{N_0}{N} + NKC \log_2 D + KD \log_2 \frac{T^L}{D}}{P} .$$

*Remark* 4.2. One must have $R(F) \leq R^{\mathsf{max}}(F)$, and equality is obtained if the model is *perfect*. For a fixed dataset, further increases in model size do not yield additional knowledge, thus $R^{\mathsf{max}}(F)$ approaches zero as the model size $P$ increases. On the other hand, Theorem 3.1 implies, ignoring lower-order terms, that if the model parameters are 8-bit (such as int8), then $R(F) \leq 8$.

---

[11]We use $\mathbb{E}_n$ or $\mathbb{E}_{n,a}$ to denote uniform random selection of $n \in \mathcal{N}, a \in \mathcal{A}$.

For our $\mathsf{bioS}(N)$ data, we define a slightly reduced capacity ratio by omitting the diversity term.[12]

**Definition 4.3.** *Given a model $F$ with $P$ parameters trained over the $\mathsf{bioS}(N)$ dataset $\mathcal{Z}$, suppose it gives $p_1 = \mathsf{loss}_{name}(\mathcal{Z})$ and $p_2 = \mathsf{loss}_{value}(\mathcal{Z})$, its capacity ratio*[13]

$$R(F) \overset{\text{def}}{=} \frac{N \log_2 \frac{N_0}{e^{p_1}} + N \log_2 \frac{S_0}{e^{p_2}}}{P} \quad and \quad R^{\mathsf{max}}(F) \overset{\text{def}}{=} \frac{N \log_2 \frac{N_0}{N} + N \log_2 S_0}{P}$$

*for $N_0 = 400 \times 400 \times 1000$ and $S_0 = 2 \times (12 \cdot 28 \cdot 200) \times 200 \times 300 \times 100 \times 263$ (c.f. Footnote 10).*

*Remark* 4.4. Ignoring names, each person contains $\log_2(S_0) \approx 47.6$ bits of knowledge.

## 5 MAIN BODY OF THIS PAPER

This paper contains too many technical results (12 in total), so many that it was initially rejected by NeurIPS 2024 since the paper "is very valuable to the developers and researchers of GPT-like LLMs" but "deserves a full hour to present than a focused conference paper (which only has a few minutes to present)", quoting the original words from the NeurIPS area chair (AC).

For this reason, we omit all technical details in this ICLR 2025 camera-ready version and encourage readers to refer to our full paper at `ssrn.com/abstract=5250617`, or to our ICML 2024 tutorial at `youtu.be/yBL7J0kgldU`. We remark that the full paper underwent the ICLR 2025 review process, but we elected to present this camera-ready version as an *extended abstract*, aligning with the tradition in the theory community.

## 6 CONCLUSION

We investigated the scaling laws of language models, specifically the relationship between model size and the total bits of knowledge stored. Our findings reveal a *precise, universal* scaling law: a sufficiently-trained transformer (i.e., one whose training loss has plateau-ed) can store 2 bits of knowledge per parameter, even when quantized to int8, which is only $1/4$ away from the information-theoretical maximum. We also examined how these scaling laws are influenced by various hyperparameters, including training duration, model architectures, floating-point precision, sparsity constraints like MoE, and data signal-noise ratios.

In terms of knowledge capacity, our methodology provides a **more accurate and principled playground** for comparing model architectures, training techniques, and data quality. We believe this playground can assist practitioners in making informed decisions about model selection, training data preparation, and further theoretical research into LLMs. Finally, our research represents an initial step towards addressing a fundamental question: how large does a language model need to be? We hope our findings will inspire further research in this area. Ultimately, we aim to provide a principled answer to the question, "Are language models with 1T parameters sufficient to achieve AGI?" in the future.

Finally, Part 3 of this work series focuses on how language models store, extract and manipulate knowledge (including Part 3.1 and 3.2 (Allen-Zhu & Li, 2024; 2025)). We also cover grade-school math and reasoning in Part 2 (Ye et al., 2025a;b), hierarchial language structure learning in Part 1 (Allen-Zhu & Li, 2023), and architecture design in Part 4 (Allen-Zhu, 2025).

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
