# OpenReview forum: "Physics of Language Models: Part 3.3, Knowledge Capacity Scaling Laws"
_ICLR.cc/2025/Conference — ICLR 2025 Spotlight_

### Official Review · Reviewer_aRy5 · 2024-10-26

**Soundness:** 4
**Presentation:** 4
**Contribution:** 4
**Rating:** 10
**Confidence:** 4

**Summary:**

This paper studies knowledge capacity scaling laws for language models. The authors design a set of novel and solid evaluation methods to assess the knowledge capacity ratio of language models. The evaluation covers different model architectures from dense to sparse LLms as well as different settings such as quantization and data noise, making it a complete and solid paper.

**Strengths:**

1. The proposed knowledge capacity evaluation is novel and useful in practical applications
2. The experiments are solid and comprehensive

**Weaknesses:**

One limitation could be the readability of this paper. Some figures are not visible such as result 8. The fonts in some figures such as Figure 1 and 3 are also too small to read. I would suggest the authors remove some sections such as result 8 to appendix, therefore more space could be left for more detailed explanation on other sections.

**Questions:**

Current I do not have questions on this paper.

---

> ### Author Response · Authors · 2024-11-28
> **Response to Reviewer aRy5**
>
> We are very grateful for the reviewer’s support on this work. Indeed we tried our best to make the experiments as comprehensive as possible, thereby putting too many results into the main body. Thank you. We can take the suggestion to remove a result from the main body (such as the quantization part).

---

### Official Review · Reviewer_e6BX · 2024-11-03

**Soundness:** 3
**Presentation:** 3
**Contribution:** 3
**Rating:** 8
**Confidence:** 4

**Summary:**

This paper examines scaling laws concerning model size versus its knowledge storage capacity. The paper addresses the following research questions:
- Does the knowledge scale linearly with  model size, and what is the exact constant of this scaling?
- How does training affect model capacity?
- How does  model architecture relate to model capacity?
- How do quantization and model sparsity affect model capacity?
- How does irrelevant/noisy data affect model capacity?

Unlike prior studies that evaluate a model’s capability via loss
or benchmarks, this paper  estimates  the number of knowledge bits a model store, focusing  on factual knowledge represented as tuples, such
as (USA, capital, Washington D.C.) from a Wikipedia page. Experimental results across model architectures establish that language models can only store 2
bits of knowledge per parameter. Detailed analyses further illustrated that a sufficiently trained 7B language model can store 14B bits of knowledge. Achieving 2bits per parameter capacity requires each knowledge piece to be revisited 1000 times. Quantizing to int8 does not compromise model capacity,  however, quantizing to int4 reduces capacity to 0.7bit/param.
Mixture-of-experts (MoE)  only reduce 1.3x in capacity,  despite using just 8.8% of the total parameters during inference.
Finally, noisy data significantly reduces model capacity but an  effective mitigation is to prepend a special token to all useful knowledge.
The model autonomously identifies high-quality data without prior knowledge of
valuable domains.

**Strengths:**

- Scaling laws are of interest to the community as they allow us to analyze and understand the capacity of large language models. They also enable the design of new models and pertaining experiments. The authors claim they are the first to propose a scaling law for the knowledge capacity of LLMs.

- The paper contains a very thorough and exhaustive list of experiments, answering several questions.

- There are practical recommendations about LLM model builders, such as domain tagging for pertaining data.

- the paper provides an explanation as to why Quantized models perform on part to their non-quantized variants in terms of knowledge storage, and similarly why  mixture of experts models perform decently despite being sparse.

- Finally, it reveals specific architectural choices in models like Llama and Mistral which might lead to slightly inferior performance (e.g., weight tying, MLP layers).

- The experimental framework is reproducible, and although the authors focus on binary knowledge base tuples, it could be extended to of the knowledge-based facts or event language structures.

**Weaknesses:**

- the paper is very dense, the appendix is very many pages long.  As a result, it is not easy to absorb all important details.

- the graphs are in very small scale and are not explained appropriately.

**Questions:**

- Please explain why you selected GPT2, Llama, and Mistral model families for your experiments. Although I understand that these represent the most popular model families, it would have been interesting to see how other, less similar architectures fare like S4.

- I would have been good to mention something about the computational requirements for your experiments.

- Aside from knowledge-base tuples, are there any other areas you think your framework might be applied to?

- I would have liked to see explicit suggestions to researchers pertaining or using LLMs. It seems that you are saying that the
quality of the data and the times of exposure matter a lot more than differences in architecture.

---

> ### Author Response · Authors · 2024-11-28
> **Response to Reviewer e6BX**
>
> We sincerely thank the reviewer for liking this work. Let us answer the specific questions.
>
> * We have chosen GPT2/Llama/Mistral as they are popular and mature decoder-only models. We have avoided encoder-only models due to their “incapability” to store knowledge properly (see cited prior work 2309.14316). We have avoided SSM models including S4 or Mamba, because we think their knowledge capacity may also have some dependency (or even trade-off) relating to the supported context length, so may be worth a totally new and thorough study.
> ** In case you’re interested in the details, let’s take the original Mamba architecture as an example. The trainable weight matrix in (each of) its SSM layer is (3D) x D, while in MLP this is (4D) x D. Therefore the recurrent state is of dimension O(D^2), where in full attention the model can read L*D floats (from the previous L tokens each of dimension D). To some extent, this makes Mamba perhaps only useful compared to full attention models when context length L >> D. In this parameter regime, one not only wants to study short knowledge sentences (such as “Joe Biden was born on Nov 20, 1942” but also long ones such as “Joe Biden …. (2000 tokens) by the way his birthday is Nov 20, 1942”. This can make the investigation significantly more involved in this paper, so we think it deserves a future work.
>
> * As for computational requirements, may I know what you’re asking specifically? We have given details of how the number of exposures translate to # of training tokens in the appendix for each dataset. In the caption of Figure 1, we briefly mentioned if using A100x64, you need 8.5 days to pretrain our largest model for 1000 exposures. This shall of course vary if you have better and more GPUs.
>
> * As for other areas this framework might apply to, one immediate application is regarding long-context knowledge tuples (such as to study S4 as discussed above). One much more involved direction is to study knowledge manipulation capabilities (such as A->B and B->C, so A->C) more thoroughly, such as using two bipartite knowledge graphs and check what’s the network needed to have them combined. We are exploring some follow-ups in such directions.
>
> * As for explicit suggestions, apart from adding domain tokens, as you said improving data quality can significantly reduce the required model size (and this is one of the main reasons of success behind the Phi-3 work, and Phi-3 is directly motivated by this submitted work). We are exploring other implications, and shall report that when it is ready.

---

### Official Review · Reviewer_ZiZe · 2024-11-03

**Soundness:** 3
**Presentation:** 2
**Contribution:** 3
**Rating:** 6
**Confidence:** 3

**Summary:**

The authors investigate how the size of language models influences their ability to store factual knowledge. Unlike previous studies that assess model capabilities through loss metrics or benchmarks, this research estimates the amount of knowledge from an information theory perspective. The authors find that an LLM can store about 2 bits of factual knowledge per parameter, meaning a 7B parameter model can hold approximately 14B bits of knowledge, covering Wikipedia-level information.
Other influencing factors of the amount of stored information include: extended training, specific architectures (e.g., GPT-2 with rotary embeddings), and higher data quality. Precision reduction (e.g., int8) minimally impacts storage, while domain annotations (like labeling Wikipedia data) significantly boost capacity.

**Strengths:**

1. The authors propose a novel method to measure the knowledge stored in an LLM by a bit complexity lower bound, and show that the amount of information stored in a single parameter is approximately 2 bits.

2. The authors investigate various influencing factors to the knowledge storage capacity of LLMs, which provide many practical insights to LLM training.

**Weaknesses:**

1. What do the long plateaus in the stored information mean? Does the maximum information reached by the plateaus equal to the bit complexity upper bound? The main scaling law only describes the linear increasing part, not the plateaus part.

2. The scaling law seems to be incomplete. i.e. the paper describes the influencing factors to the stored knowledge separately. Is there an empirical law that can describe and summarize all the results?

3. What does it mean if we say that the LLM stores N bits of knowledge? Does it mean this amount of knowledge is ready for extraction or just memorized in a fixed form? There are some memorization v.s. extraction accuracy plots in the appendix, which seem also show a first linearly increase and then plateau trend, but these results are not well associated with the bit information results. e.g. Would the bits of knowledge stored linearly be associated with the knowledge memorization/extraction accuracy? If the meaning of this information theoretical definition of storage knowledge is not well understood, it is hard to interpret the insights obtained from it.

4. The derivation of the bit complexity lower bound feels a bit ad hoc. It seems that the authors just tried to build a lower bound in a similar form to the upper bound involving the training losses. Some steps in the derivation do not feel natural to me, especially lemma 1.6. What is the meaning of the RHS of equation I.1? How tight would this lower bound be?

5. The training data generation process is basically uniformly random. There are also similar uniformly random assumptions used in the proof of the bit information lower bound. I wonder if the derived lower bound can be applied to real-world data, which is not uniform, but usually naturally exhibits a long-tailed distribution. I understand that certain assumptions are needed for synthetic experiments, but it is also important to understand how much the obtained conclusions can be applied to real-world applications.

6. The writing can be improved.

Overall, I appreciate the information theory approach adopted and the empirical results seem to be quite inspiring.
I'm willing to improve my score if my concerns are properly addressed.

**Questions:**

See weaknesses.

---

> ### Author Response · Authors · 2024-11-28
> **Response to Reviewer ZiZe**
>
> We thank the reviewer for the time and efforts on reading this paper and raising specific concerns. We address them one by one.
>
> ## Response to question 1.
>
> **For the long plateaus**, yes they mean the model has properly stored all the knowledge bits from the data. We wrote this in Result 1(b) in Line 328. In words, all models (not just a cherry-picked set of models) that exceed a size limit shall be able to attain perfect knowledge accuracies.
>
> ## Response to question 2.
>
> We argue that there may not be **any single empirical law**. As we showed in the “Junk Data vs Scaling Law” section, the richness/junkness of data can significantly influence the model’s knowledge capacity. As a result, models trained using high quality data (such as Phi-3) can store many more knowledge bits comparing to models trained directly on common crawl. An “empirical law” would therefore heavily depend on the data type. On the other hand, using controlled data can allow us to study the influencing factors directly, thus giving more empirical suggestions on how to prepare the data and so on.
>
> ## Response to question 3.
>
> As for “storing knowledge” vs “**extracting** knowledge”, yes storing knowledge (token-by-token) doesn’t always imply the knowledge can be flexibly extracted for downstream use. This was pointed out in cited prior work 2309.14316. In short, this submission has made sure that the counted knowledge is flexibly extractible (not just token-level memorization).
>
> Longer story, when you compare Figure 7 with Figure 1, where we checked the accuracy on extracting knowledge (such as “What’s the birthday of <name>”, following the same definition of knowledge extraction from prior work 2309.14316). We argue that **Figure 7 is almost identical to Figure 1**. Please note, there’s roughly a factor 50 difference in the y-axis between Fig 1 and 7, because the knowledge (birthdate/city/etc.) of each person has 47.6 bits of knowledge. Therefore, the 2bit/param curve translates to 0.042person/param or equivalently 24 param/person.
>
> In particular, you still have an (almost) linear capacity law **in Figure 7 and the plateau is for the same reason as Figure 1** — model has already 100% answered all the knowledge during extraction.
>
> A minor remark: when computing knowledge extraction accuracy in Figure 7, we can only plot the N * log (S0) part which is the total knowledge of all the people. We haven’t included the knowledge bits for memorizing the person names — the N * log (N0/N) part. This is not captured by the knowledge extraction accuracy, and explains why in Figure 7, when N is small, the curve is a little off from the linear line, because it hasn’t included the person name part.
>
> ## Response to question 4.
>
> As for the **derivation of the bit-complexity lower bound**, the fact that it looks in a similar form to the upper bound, this is just the outcome of the math. When the training loss is zero this bound is of course very tight.
>
> When it is not, to give you a simplest example, say there are N=2 people each having a single-token value inside 1..V. If ignoring the person names, this is 2 * log2(V) bits of knowledge (upper bound). Say now a model has loss p1 and p2 on predicting the two people’s values respectively. Then it “must have” memorized log2(V/e^p1) + log2(V/e^p2) bits of knowledge (lower bound). This is the same as 2 * log2(V/e^p) for p = (p1+p2)/2 which is the expected loss. So this lower bound is also tight in this case, and is of the same form as the upper bound.
>
> Please note, while this lower-bound argument looks natural, to make it rigorous — namely to explain what “must have” means — one needs a math proof. This can be found in Line 1710-1725, the first "value-only" warmup case.
>
> What makes the lower bound not very tight is the additive o(KD) term, which will show up only when the person names are also involved, but is anyways a small term.
>
> ## Response to question 5.
>
> As for the concern on the **uniform randomness**, please note there are two such uniformities.
>
> One is on the distribution of value (for instance a person’s birth city is uniformly chosen from 100 choices). This is solely for the ease of presentation. If values are uniform this is log2(100) bits; if not, the bit depends on the distribution so the formula is ugly, but the results largely stay the same.
>
> The more interesting uniformity is that we have chosen all knowledge to have the same 100/1000 exposure. While it appears more interesting to study “power law” distribution, please note our “Junk Data vs Scaling Law” section precisely tackles this. When data are of different qualities, the knowledge capacity changes. We believe our controlled setting of having exactly 2 data frequencies can explain the influence of “junk data” with a better contrast, as opposed to uniform vs power law distribution.

---

> > ### Comment · Reviewer_ZiZe · 2024-12-01
> >
> > I thank the authors for their response. It's very helpful and resolved a large part of my concerns so I will raise my score to 6. On the other hand, I have some follow-up comments:
> >
> > 1. To my understanding, result 1(b) describes the maximum information **ratio** (R^max), or the slope of the increasing part in the figure, not the plateaus -- maximum information **bits**. From the figure, it seems to be N-specific. Shouldn't there be some description of the value of this maximum (perhaps as described in Proposition 2.3) and when this maximum shall be reached? This might have practical insight into how large an LLM is needed to store all the (storable) information of a dataset.
> >
> > 2. Although there might not be a single empirical knowledge scaling law in the real world, it is still possible to obtain a somewhat complete scaling law in the current synthetic setting. But I can see that the current incomplete form already has many interesting insights so I think it's ok to delay this to future work.
> >
> > 3. I suggest the authors include this argument about the connection between stored knowledge and extractable knowledge in the paper. It would be even better to show empirical/statistical relations between Figure 1 and Figure 7. e.g. Through fitting a linear regression.
> >
> > 4. I suggest the authors include this intuitive example in the main paper so that readers can easily understand the intuition/meaning of the lower bound.
> >
> > 5. I don't think power-law distributed data implies the data is of low quality or even means it's junk data. The “Junk Data vs Scaling Law” section provides some insights into the importance of data quality, but it is inevitably still an oversimplification. The success of LLM is driven by scaling, which includes the scaling of pretraining data. Under this scale, the data is naturally power-law distributed. The current knowledge scaling law has its own merit under the uniform distribution setting, while it is still interesting/necessary to understand knowledge scaling in a real-world pretraining setting. I suggest the authors add a limitation section to discuss these issues.

---

> > > ### Author Response · Authors · 2024-12-03
> > >
> > > Thanks a lot for your re-consideration! We appreciate your follow-up questions and let us answer them in details.
> > >
> > > > 1. To my understanding, result 1(b) describes the maximum information ratio (R^max), or the slope of the increasing part in the figure, not the plateaus -- maximum information bits. From the figure, it seems to be N-specific. Shouldn't there be some description of the value of this maximum (perhaps as described in Proposition 2.3) and when this maximum shall be reached? This might have practical insight into how large an LLM is needed to store all the (storable) information of a dataset.
> > >
> > > **Our response:** Our Result 1 has three parts: **Result 1(a)** describes the 2bit/param **slope**, but we are not satisfied with this and wish to show this is **tight from above and below**.
> > > * Result 1(c) says this is "tight from above", meaning that no model can outperform 2.3bit/param
> > > * Result 1(b) says this is "tight from below", showing the **plateau**, namely all the models that have R^max<=1.8, they can achieve $R \approx R^{max}$, so they have acquired all the knowledge from the dataset. **To put it in words**, if the data has 50,000,000 bits of knowledge --- as it does for bioS(N=10m) --- then all models that have more than 25 million parameters can store nearly perefectly all the 50M knowledge bits.
> > >
> > > The plateau is indeed N-specific, if you have 10x more people's biographies you roughly have 10x more bits of knowledge. As you suggested, this **Result 1(b) indeed gives practical insight about what's the minimum size an LLM needs to be** in order to store N bits of knowledge -- the answer is N/2 parameters if using 1000 exposures of perfect data.
> > >
> > > ***
> > >
> > > > 2. Although there might not be a single empirical knowledge scaling law in the real world, it is still possible to obtain a somewhat complete scaling law in the current synthetic setting. But I can see that the current incomplete form already has many interesting insights so I think it's ok to delay this to future work.
> > >
> > > **Our response:** In fact that's where we started. Originally we have tried different power-law distributions (to mimic real world) but observed that the model has different scaling laws, heavily dependent on the power-law parameter. When a uniform exposure is used, this issue goes away. This is why we ended up using a cleaner "junk vs knowledge" comparison to explain this, and at the same time showcase the harm of "junk" data.
> > >
> > > (Also answer your question 5), if using power-law distribution, say for instance if you take all the knowledge that have appeared >=100 exposures and count that's N bits, then you may need 5N-sized models to learn all of them, due to the existence of all the knowledge that has appeared <=20 or even fewer number of exposures. In other words, you will see similar "capacity degrade" for power-law distribution, because of the existence of such "junk data" (e.g., with <=20 exposures and less)
> > >
> > > ***
> > >
> > > > 3. I suggest the authors include this argument about the connection between stored knowledge and extractable knowledge in the paper. It would be even better to show empirical/statistical relations between Figure 1 and Figure 7. e.g. Through fitting a linear regression.
> > >
> > > **Our response:** Hmm, isn't Figure 7a vs 7b, and 7c vs 7d exactly what you want? The left two figures (7a and 7c) are for memoizable knowledge (for definition see Line 1009-1014), and the (7b and 7d) are for extractable knowledge. Short answer is that they are almost identical.
> > >
> > > ***
> > >
> > > > 4. I suggest the authors include this intuitive example in the main paper so that readers can easily understand the intuition/meaning of the lower bound.
> > >
> > > **Our response:**  Thanks! will do

---

### Official Review · Reviewer_Z8yV · 2024-11-04

**Soundness:** 3
**Presentation:** 2
**Contribution:** 3
**Rating:** 5
**Confidence:** 3

**Summary:**

The paper investigates the relationship between language model size and its capacity to store factual knowledge, quantified in bits per parameter. The authors introduce a framework that measures a model’s knowledge based on tuple-based information (e.g., (Entity, Relation, Attribute)) and propose that language models, after sufficient training, achieve an approximate capacity of 2 bits per parameter. The study extends this analysis across various factors such as model architecture, quantization levels, sparsity, and training data quality. Using synthetic and controlled datasets, the authors examine how these elements influence the knowledge storage capacity of language models.

The technical claims are generally supported by experiments; however, there are concerns about the robustness of the findings. The reliance on synthetic data raises questions about the applicability of the results to real-world scenarios. Additionally, the paper does not thoroughly explore the impact of quantization during training.

The paper has several formatting issues that hinder its readability. Notably, it lacks a conclusion section. Some figures are not clear. The organization of the paper could be improved drastically.

**Strengths:**

•	Originality: Introducing a framework to measure language model capacity in bits per parameter is a novel approach that adds a quantitative dimension to model evaluation.

•	Methodology: The use of controlled synthetic datasets allows for the isolation of specific variables, providing clarity in the analysis of different factors affecting knowledge capacity.

**Weaknesses:**

•	Formatting Issues: The absence of a conclusion section and unclear figures detract from the overall quality of the paper and impede the reader’s understanding.

•	Generalization to Real-world Data: The heavy reliance on synthetic data limits the applicability of the findings to natural language processing tasks involving complex and diverse datasets.

•	Incomplete Exploration of Quantization: The paper does not investigate quantization during training, which could provide insights into mitigating the observed decrease in capacity with int4 quantization.

•	Limited Architectural Diversity: The study does not explore a wide range of model architectures, such as encoder-only or decoder-only models, which could affect the generalizability of the proposed scaling law.

**Questions:**

1.	How would the proposed 2-bit/parameter scaling law hold up when applied to language models trained on real-world, diverse datasets?

2.	Could incorporating quantization into the training process mitigate the reduction in capacity observed with int4 quantization?

---

> ### Author Response · Authors · 2024-11-28
> **Response to Reviewer Z8yV**
>
> We sincerely thank the reviewer for appreciating our originality and methodology. Indeed, the use of controlled data can make us better examine the effect of different factors affecting knowledge capacity.
>
> As for connection to **real-world** data:
> * Our “_Junk Data vs Scaling Laws_” showed that, if you train using real-world data (esp. common crawl), you don’t get 2bit/param.
> * We actually first discovered this when we pretrain on *real-world* WikiBook + Bio data, and then decided to make the experiment more “controlled” by removing WikiBook but instead using a controlled Junk data. This enabled us to study the relationship between knowledge density/junkness vs model capacity.
> * This controlled setting also helped argue for potential changes needed when pretraining with real-life data (such as adding domain tokens).
>
> As for **quantization**, yes it is interesting to study quantization-aware training methods. Since there are many different such methods and subtleties, and since this submission already contains too many results, we decide to leave it a future work. In this way, the main focus of this submitted paper is this knowledge-bit methodology, and we demonstrated that it can be useful in many areas (including studying quantization).
>
> As for formatting suggestions:
> * indeed due to space limitation, we greatly reduced the length of our conclusion and it contains only 8 lines (at the end of Section 1).
> * indeed due to the novel methodology, the figures we show are unconventional (not just accuracy, but bit complexity and exposure) and require careful reading and explanation. Please feel free to let us know if you have concrete suggestions on how to improve the figures.
>
>
> Thanks again for your time!

---

### Meta-Review · Area_Chair_v55H · 2024-12-25

**Metareview:**

This paper investigates the knowledge storage capacity of language models, revealing that models can store approximately 2 bits of factual knowledge per parameter. The study explores how factors like model architecture, training duration, quantization, sparsity, and data quality influence knowledge storage.
Strength: The paper introduces a novel framework to measure language model knowledge capacity in bits per parameter. Examining the knowledge capacity scaling also provides an important new direction to understand LLMs. The thorough and comprehensive experiments provide practical insights for LLM training. Additionally, the work develops practical recommendations, such as domain tagging and the importance of using high-quality data.
Weakness: As pointed out by reviewers, the paper’s study mainly relies on synthetic data, which can effectively isolate variables in real data. However, it could also potentially limit its applicability to real-world scenarios with complex and diverse datasets. Reviewers also pointed out that the content of the paper is quite dense with some unclear figures, which should be further improved in the revised version.

Overall, this paper provides a significant contribution to the area of LLMs, especially in its understanding of the knowledge learning and capacity scaling. The authors have sufficiently addressed majority of the reviewers’ feedback.

**Additional Comments On Reviewer Discussion:**

Majority of the feedbacks from the reviewers have been addressed and clarified by the authors. Although studying the problem with synthetic data may impose potential weakness, it also provide a good benefit of studying the problem in a fully controlled manner. The insights and observations obtained from such a methodology is also valuable in its own. Also, the authors should take the reviewer feedback on further improving the presentation of the paper.

---

### Decision · Program_Chairs · 2025-01-22

Accept (Spotlight)